# Nutritional Interactions between Bacterial Species Colonising the Human Nasal Cavity: Current Knowledge and Future Prospects

**DOI:** 10.3390/metabo12060489

**Published:** 2022-05-27

**Authors:** Lea A. Adolf, Simon Heilbronner

**Affiliations:** 1Interfaculty Institute for Microbiology and Infection Medicine, Institute for Medical Microbiology and Hygiene, UKT Tübingen, 72076 Tübingen, Germany; lea.adolf@student.uni-tuebingen.de; 2German Centre for Infection Research (DZIF), Partner Site Tübingen, 72076 Tübingen, Germany; 3Cluster of Excellence EXC 2124 Controlling Microbes to Fight Infections, 72076 Tübingen, Germany

**Keywords:** nasal microbiome, *Staphylococcus aureus*, bacterial interaction, nutritional interaction

## Abstract

The human nasal microbiome can be a reservoir for several pathogens, including *Staphylococcus aureus*. However, certain harmless nasal commensals can interfere with pathogen colonisation, an ability that could be exploited to prevent infection. Although attractive as a prophylactic strategy, manipulation of nasal microbiomes to prevent pathogen colonisation requires a better understanding of the molecular mechanisms of interaction that occur between nasal commensals as well as between commensals and pathogens. Our knowledge concerning the mechanisms of pathogen exclusion and how stable community structures are established is patchy and incomplete. Nutrients are scarce in nasal cavities, which makes competitive or mutualistic traits in nutrient acquisition very likely. In this review, we focus on nutritional interactions that have been shown to or might occur between nasal microbiome members. We summarise concepts of nutrient release from complex host molecules and host cells as well as of intracommunity exchange of energy-rich fermentation products and siderophores. Finally, we discuss the potential of genome-based metabolic models to predict complex nutritional interactions between members of the nasal microbiome.

## 1. Introduction

Human body surfaces are colonised by a multitude of different microorganisms from all three kingdoms of life as well as their viruses [1]. Bacteria and bacteriophages, however, represent the most abundant members of this ensemble, which is referred to as the microbiome [2]. A growing body of epidemiological, clinical and experimental studies indicate that microbiomes modulate fundamental physiological processes in humans and that dysbiosis of the microbiota is associated with several acute and chronic disorders, including psoriasis, asthma, obesity and cardiovascular diseases, as well as infection [3]. The increased risk of infection associated with changes in the composition of microbiomes is related to their function as reservoirs of bacteria that can become pathogenic.

The human nasal microbiome is reservoir of several important human pathogens. In children, pathogenic species such as *Neisseria meningitidis*, *Haemophilus influenzae* and *Streptococcus pneumoniae* are frequently found [4]. In adults, the most prominent pathogen is *Staphylococcus aureus*, which asymptomatically colonises the anterior nares of about one-third of the human population. *Staph. aureus* causes a wide range of invasive infections, leading to high morbidity and mortality, and asymptomatic nasal colonisation represents a major risk factor for infection [5]. Additionally, the nasal mucosa is a portal of entry of pathogenic viruses, fungi and other bacteria. The interplay between a healthy nasal microbiome and the host immune system provides the first barrier against these agents [4,6,7]. Moreover, a healthy microbiome is key to preventing hyperreactivity of the immune system of the upper and low respiratory tract. Consequently, nasal microbiome dysbiosis is often associated with the exaggerated immune responses underlying respiratory disorders such as rhinosinusitis, asthma and chronic obstructive pulmonary disease, which in turn increase the risk of chronic respiratory infections [4,8,9,10].

The composition of the human nasal microbiome changes dramatically over the life span [4] but differences between individuals are also dramatic. In healthy human volunteers, seven different “community state types (CST)” have been identified, only one of which is dominated by *Staph. aureus* [11]. Host genetics plays a minor role in determining the presence or absence of *Staph. aureus.* This strongly suggests that the community structure and the interactions between *Staph. aureus* and other nasal commensals are decisive in either allowing or preventing *Staph. aureus* colonisation. This opens up the possibility to manipulate the nasal microbiomes in order to prevent infection. However, this requires a better understanding of the molecular interactions that occur between the members of the nasal microbiome in the environmental conditions present in the human nasal cavity. Unfortunately, this knowledge is fragmented. Research on nasal colonisation by *Staph. aureus* has focused on the pathogen and host, while the contribution of the nasal microbiota is poorly understood.

Additionally, the general principles behind microbe–microbe interactions have mainly been inferred from knowledge of the gut microbiome. The dynamics of microbiomes are shaped by multiple factors including the environment (e.g., temperature, pH, oxygen), nutritional status and host immune responses. However, there is little similarity in these parameters between different human body sites [12]. The environment within the human gut is characterised by anaerobic conditions and the presence of high amounts of complex nutrients. Furthermore, peristaltic movement promotes constant mixing of the microbiome. In contrast, external body surfaces such as the nares are predominantly aerobic; they have low levels of nutrients and there is less mixing of the microbiome [13]. This suggests that interspecies competition or collaboration within the nose microbiome will differ from those that occur in the gut.

Beneficial microbes contribute to resistance against opportunistic pathogens through direct or indirect mechanisms. Indirect effects are largely mediated by microbial stimulation of host immune responses [14]. Direct effects include killing by antibiotics [15], contact-dependent killing by Gram-negative bacterial type VI secretion systems [16], adhesion exclusion and signaling interference [17]. Such mechanisms have been extensively reviewed elsewhere and will not be discussed here.

Competition for or collaborative efforts to acquire scarce nutrients are less well-understood. In this review, we will focus on the nutritional interactions that are known to occur between the members of the human nasal microbiome. We are convinced that these interactions are of special relevance because the nasal microenvironment is characterised by scarce availability of nutrients. As knowledge derived from the nasal microbiome is limited, we highlight general concepts of nutritional interactions that have been observed in microbiomes from other human body sites, including the skin, the mouth, the respiratory tract and the gut, and discuss their potential relevance for the human nasal community.

## 2. Nutritional Interactions amongst Species of the Nasal Microbiome

The human nasal cavity is poor in energy-rich, low-molecular-weight nutrients such as mono- or disaccharides that can be directly acquired by microbes. Similarly, nitrogen-containing nutrients such as amino acids are scarce, and the availability of trace metals is limited [13,18,19]. This promotes two distinct patterns of interactions between bacteria that inhabit the nares. Firstly, individuals cooperate in using limited resources. For example, energy-rich fermentation products of one organism can be consumed by others. In addition, the secretion of hydrolytic enzymes by one organism can release nutrients from high-molecular-weight molecules for the benefit of the entire community. Secondly, nutritional limitation will promote competition. This effect will be especially relevant for closely related bacteria with similar metabolisms and similar nutritional requirements [20,21,22]. Competition will occur also between distant bacterial lineages when essential trace nutrients such as transition metals are limited.

### 2.1. Secreted Small Molecules

The metabolic activity of bacterial cells and their secreted products can influence the characteristics of their habitats. This can open ecological niches for other microbes and foster interactions. These interactions can be mutualistic or inhibitory and contribute to shape the structure of bacterial communities.

#### 2.1.1. Energy-Rich Fermentation Products

Bacterial fermentation of carbohydrates is accompanied by the secretion of energy- rich molecules such as short-chain fatty acids (SCFAs), lactate, succinate, formate or alcohols. While these products represent end products for the producer, they can be used as substrates by cocolonising species with different metabolic capacities. Most fermentation products are toxic at higher concentrations so metabolic interactions can be of reciprocal benefit for producer and consumer. While the consumer benefits from the energy content of the fermentation products, the producer profits from the associated detoxification of the environment (Figure 1).

Cross-feeding interactions have been studied in several bacterial habitats. Regarding those associated with humans, several interactions have been reported in the context of the gut. For example, *Bifidobacterium adolescentis* secretes lactate, which can be used by *Eubacteirum hallii* and *Anaerostipes caccae* [23]. Similarly, *Lactobacillus* spp. and *Bifidobacterium longum* are able to degrade inulin-type fructans followed by secretion of lactate and acetate, which support butyrate-producing colon bacteria [24].

To our knowledge, the metabolic interactions between the members of the human nasal cavity have not been studied. However, some examples are available in the context of cystic fibrosis (CF). *Pseudomonas aeruginosa* and *Staph. aureus* are important pathogens in CF patients, and the two organisms are frequently found in mixed communities [25,26,27]. Camus and colleagues have recently demonstrated that the two species can reciprocally support each other [28]. During fermentation *Staph. aureus* secretes toxic acetoin. In coculture, *P. aeruginosa* was shown to use acetoin as an alternative carbon source when the concentration of glucose was low. This interaction is mutually beneficial because acetoin increases *P. aeruginosa* proliferation while simultaneously detoxifying the environment to support *Staph. aureus* survival [28].

In addition, the virulence of *P. aeruginosa* is increased by cohabitant bacteria such as *Klebsiella aerogenes*, *K. pneumoniae*, and *Staph. aureus* in CF patients. They produce the fermentation metabolite 2,3-butanediol, which promotes biofilm formation by *P. aeruginosa*. It also increases colonisation of the respiratory tract by environmental microbes [29,30]. As 2,3-butanediol can be transformed into acetoin [31], it might serve as a carbon source for *P. aeruginosa*.

Additionally, mucin-degrading bacterial species and their fermentation products are important for structuring communities, which is discussed in detail below.

#### 2.1.2. Siderophores

Another important nutrient is iron. Under physiological conditions, ferric iron (Fe^3+^) is the dominant state. Ferric iron is hardly soluble, which limits its bioavailability. This is intensified on host mucosal surfaces by the secretion of the iron-binding molecule lactoferrin. To overcome iron limitation, bacteria secrete siderophores. Siderophores have a very high affinity for Fe^3+^. The iron-complexed forms are taken up by bacterial cells via ABC transporters, as reviewed elsewhere [32]. In general, iron-saturated siderophores can be acquired by all members of a community since siderophore acquisition solely depends on the expression of an appropriate receptor [33,34,35] (Figure 1). Siderophore production is metabolically costly and results in the emergence of “cheaters” that rely on siderophore production by other strains. The presence of “cheaters” negatively affects the fitness of producers and has important effects on bacterial populations. This has been extensively reviewed elsewhere [33].

Siderophore production and cheating might also influence the structure of nasal communities. Nasal secretions are iron-limited, and iron-acquisition genes are strongly expressed by *Staph. aureus* during nasal colonisation of humans and experimental animals [18,19]. *Staph. aureus* produces two siderophores: staphyloferrin A and staphyloferrin B. The iron-saturated forms are taken up by the Hts and Sir systems, respectively. Additionally, *Staph. aureus* encodes systems for the acquisition of xenosiderophores produced by other bacteria and the human host. The FhuCBG and SstABC systems promote acquisition of hydroxamate-type and catecholate-type siderophores, respectively [36,37]. This suggests that *Staph. aureus* might profit from the presence of nasal commensals that produce these siderophores, which is a common trait of many bacteria.

The nasal commensal *Staph. lugdunensis* is unable to produce siderophores but is able to take up staphyloferrin A and B produced by *Staph. aureus*. Accordingly, the presence of *Staph. aureus* enabled *Staph. lugdunensis* to thrive under iron-restricted conditions. However, it is unclear if this phenomenon is relevant during nasal colonisation [38]. *C. propinquum* was shown to produce the siderophore dehydroxynocardamine in the human nasal environment. This resulted in the inhibition of coagulase-negative staphylococci (CoNS), suggesting that these species could not acquire dehydroxynocardamine and were therefore faced with increased iron restriction [39].

#### 2.1.3. Oxygen Consumption

Several anaerobic species are found within the nasal cavity [40] strongly suggesting the presence of anaerobic microniches. It seems possible that these niches are provided, at least in part, by physical characteristics of the mucosal surface such as crypts [41], which create anoxic areas. However, it is also likely that oxygen consumption by aerobes allows proliferation of anaerobes in close proximity (Figure 1). A relevant example is sinusitis where the abundance of anaerobic bacteria such as *Fusobacterium*, *Prevotella*, *Porphyromonas* and *Peptostreptococcus* spp. increases over time. It was hypothesised that the causes of these changes were a) the human host by creating edema and swelling, and b) the aerobic bacteria by consuming oxygen and thereby creating favourable conditions for anaerobes to grow [42].

### 2.2. Host Cells as a Source of Nutrients

The human nasal cavity is nutritionally poor. However, mucosal cells display and secrete several high-molecular-weight glycosylated proteins such as sialylated glycans, or glycosylated proteins such as mucins [43,44,45,46]. These serve as nutrient sources for microbial communities via the secretion of degradative enzymes. Secreted enzymes represent “public goods” as their activity will create products that can be acquired by other members of the microbial community. Thus, certain commensals will profit from enzymes secreted by others, which might foster cohabitation. Additionally, host cells themselves can be regarded as a source of nutrients if they are lysed by bacteria and release their cytosolic contents.

#### 2.2.1. Host Mucins as a Source of Carbon and Sulphate

Mucins are large glycoproteins that are secreted by or are membrane-anchored on mucosal epithelial cells in the gastrointestinal, respiratory, reproductive and urinary tracts. They contribute to the host’s antimicrobial defences as part of the gel-like mucus that forms a physical barrier that can aggregate pathogens and facilitate their clearance. However, mucus can also accommodate beneficial bacteria.

Mucins are highly O-glycosylated proteins carrying complex O-glycosylation structures comprising fucose, galactose, N-acetylgalactosamine or sialic acid, with the last two sometimes being sulphated [45,46]. Mucins can serve as carbon, nitrogen and sulphate sources when they are degraded by bacterial enzymes (Figure 2). Several studies showed that mucin-degrading bacteria can support proliferation of organisms that are unable to do so. However, it is difficult to assess if these effects are direct or indirect. A direct benefit would be the consumption of released amino acids or carbohydrates, while an indirect effect would be the consumption of fermentation products derived from mucin-degrading bacteria as described above. The importance of collaborative degradation of mucins is evident.

In patients suffering from chronic rhinosinusitis the abundance of anaerobic, mucin-degrading bacteria increases. In vitro, mucin degradation products foster the proliferation of *Staph. aureus* and modulate its gene expression by promoting the transition from commensal to pathogen [47]. In vivo, the degradation of mucin was shown to be beneficial to all members of the oral microbiome [48]. In a study investigating bacteria in the nasopharynx, the growth of *N. meningitidis* was increased when *Streptococcus mitis* was present. This was attributed to the ability of *Strep. mitis* to release carbohydrates from mucins, which supported the growth of *N. meningitidis* [49].

Likewise, in the lungs of CF patients, anaerobic communities comprising *Prevotella*, *Veillonella*, *Streptococcus* and *Fusobacterium* spp. were shown to degrade mucins and to produce SCFAs as metabolic end products [50]. *P. aeruginosa* cannot utilise mucins as a carbon source when growing alone. However, the SCFAs allowed it to thrive in the community where mucins were the sole carbon source [50]. In line with this observation, *P. aeruginosa* profited from the presence of mucin-degrading bacteria in a rabbit chronic rhinosinusitis model [51].

Sulphation of terminal sugars in glycans can protect mucins from bacterial degradation. However, some bacteria encode sulphatases that remove the sulphated cap and enable degradation of the glycans [52]. This was demonstrated to be of importance within the human gut microbiome. The human commensal *Bacteroides thetaiotaomicron* expresses several sulphatases, which are essential for its ability to utilise O-glycans [53].

*P. aeruginosa* secretes a sulphatase that promotes mucin degradation, which facilitates invasion of the mucus barrier and promotes virulence in a systemic mouse infection model [54].

In conclusion, the decapping mechanism could support other bacteria in the same niche by providing access to degradable glycans. It is also possible that the sulphate groups could serve as a nutrient source. However, nothing is known about any role for sulphated mucins within the nasal microbiome.

#### 2.2.2. Host Glycans as a Source of Sialic Acid

Eukaryotic mucosal cells are decorated with sialylated glycans where sialic acid is the terminal monosaccharide (Figure 2). Moreover, mucins can also contain sialic acids. Due to their negative charge and hydrophilicity, sialic acid residues have important functions in blood cell repulsion, in glomerular filtration and in determination of the half-life of circulating glycoproteins [43,44]. Sialic acids also have important biological roles as ligands for molecules such as selectins and Siglecs (Sia-binding immunoglobulin-superfamily lectins). They also act as ligands for adhesins on the surface of pathogens [43,44]. Additionally, host immune effectors such as lactoferrin or IgA2 are sialylated [55,56,57].

Bacterial manipulation of sialylated eukaryotic receptors or secreted molecules has been recognised as an immune evasion strategy [43,58,59,60]. The secreted sialidase NanA of *Strep. pneumoniae* promotes desialylation of lactoferrin and IgA2, thereby preventing bacterial clearance from the respiratory tract [55]. In addition, the released sialic acid can provide a carbon and nitrogen source for bacteria [58]. Thus, *Strep. pneumoniae* was shown to grow on human glycoconjugates using its sialidases NanA and NanB [61]. This strongly supports the idea that the release of sialic acid from host cells can provide access to nutrients. This concept might also be important in the nutritionally poor environment of the human nasal cavity. Direct evidence that nasal commensals degrade host glycans is currently lacking. However, *Cutibacterium acnes* encodes several putative sialidases [62,63]. Similarly, *Strep. mitis* and *Strep. oralis* that colonise the oral cavity express sialidases [64]. Catabolism of sialic acid has been demonstrated for *Staph. aureus* and *Staph. lugdunensis.* The *nan* locus comprising a transporter (*nanT*), catabolic enzymes (*nanA*, *nanK*, *nanE*) and a repressor (*nanR*) was shown to be responsible [65].

It can be concluded that desialylation of host glycoproteins effects both the function of the molecules and also provides a source of nutrients. Both effects will benefit the entire community and not only the organism responsible for the activity. However, the relevance of this for the composition of microbial communities is currently unclear.

#### 2.2.3. Host Fatty Acids and Phospholipids as a Source of Carbon and Phosphorous

It is well-appreciated that bacterial lipases allow several bacterial species to use host-derived fatty acids (Figure 2). This is reviewed elsewhere [66,67,68,69]. Some evidence is available to indicate that this phenomenon is of importance to the nasal microbiome. Human nasal fluid contains all major classes of lipids, including fatty acids, glycerolipids, glycerophospholipids, sphingolipids, cholesterol and cholesteryl esters [70]. Secreted bacterial glycerophosphodiesterases can degrade glycerophosphodiesters into glycerol-3-phosphate and the respective alcohol [71], while lipases hydrolyse triacylglycerols into glycerol and fatty acids [72]. The expression of the glycerophosphodiesterase GlpQ of *Staph. aureus* is regulated by phosphate availability [73]. GlpQ is an enzyme that allows *Staph. aureus* to thrive on host phospholipids as the sole source of phosphorus [74]. Additionally, *Staph. aureus* releases fatty acids from human low-density lipoproteins using its lipase GehB. The fatty acids are incorporated into the bacterial membrane, thereby increasing the growth rate of the pathogen [75]. Similarly, the human skin and nasal commensal *C. accolens* secretes the lipase LipS1 to release fatty acids from host triacylglycerols. Interestingly, the hydrolysis of triacylglycerols produces oleic acid as a by-product, which inhibits *Strep. pneumoniae* growth in the nasopharynx [76]. The skin and nasal commensal *Staph. epidermidis* secretes a sphingomyelinase that degrades the human membrane lipid sphingomyelin into ceramide and phosphocholine. Phosphocholine contains carbon and nitrogen and is used by *Staph. epidermidis* as nutrient source. Moreover, ceramide production supports the human host by preventing skin dehydration, which is important for the skin-barrier function [77]. *C. acnes* is a commensal of the human skin and nasal cavity. It is able to hydrolyse human triglycerides [78]. However, whether this contributes to growth of *C. acnes* was not reported. Finally, *P. aeruginosa* showed reduced growth on various human fatty acids as the only carbon source in mutants lacking FadD1 and FadD2, enzymes that are used for fatty-acid degradation. FadD1 and FadD2 might be of importance in the lipid-rich environment of the CF lung [79].

It can be concluded that there is considerable evidence regarding the degradation and use of host-derived lipids and fatty acids by individual organisms, but the relevance of this at the community level has hardly been studied.

#### 2.2.4. Host Erythrocytes as a Source of Haem

Host erythrocytes are a rich source of haemoglobin, which carries four iron-containing haem molecules. Haem-iron is used by many bacterial pathogens in the iron-limited environment of the human host. This has been extensively studied in the context of invasive disease [80]. Haemoglobin is also found within the nasal cavity and promotes colonisation by *Staph. aureus* [81]. Haem acquisition from haemoglobin might also be important for the entire nasal community (Figure 2). It is currently unclear how haemoglobin reaches the nasal cavity, but it is possible that intact erythrocytes arrive through microlesions in the epithelial barrier. If this is true, bacterial haemolysins will likely benefit the entire community. *Staph. aureus*, *Staph. epidermidis* and *Staph. lugdunensis* are haemolytic [82,83,84]. Some *C. acnes* strains are β-haemolytic under anaerobic conditions [85]. Systematic analysis of haemolytic activity by other nasal commensals is currently lacking. In addition, it is unclear if all nasal commensals use haemoglobin as an iron source. However, the pathogens *N. meningitidis* [86], *H. influenzae* [87] and *P. aeruginosa* [88] express haem-acquisition systems. *Corynebacterium diphtheriae* thrives on haemoglobin [89], so it is possible that the closely related commensal *Corynebacteria* might possess a similar ability.

In addition to haem, erythrocytes are also a source of other cellular factors. Lysis of erythrocytes by *Staph. aureus* releases haemin and NAD+, which in turn supports growth of *H. influenzae* [90]. Thus, haemolysis can be regarded as collaborative nutritional trait.

Finally, inflammation followed by tissue destruction can serve as a source of nutrients such as amino acids and haem. For example in a periodontitis model, tissue inflammation caused by the pathogen *Porphyromonas gingivalis* led to tissue destruction and release of nutrients for the benefit of the local bacterial community [91].

### 2.3. Some Microbiome Members Act as Prey to Obtain Essential Nutrients

Some bacterial cells within a community can be regarded as sources of nutrients. In Gram-positive bacteria, the cell surfaces are frequently decorated with phosphate-containing wall teichoic acid (WTA) glycopolymers. The outer membrane of Gram-negative bacteria frequently displays sialylated proteins. By expressing bactericidal compounds or hydrolytic enzymes, prey bacteria release nutrients from other bacteria.

#### 2.3.1. WTA as a Source of Phosphorus

WTA molecules are glycopolymers consisting of ribitol-phosphate or glycerol-phosphate subunits that are covalently linked to the peptidoglycan of the cell wall. These polymeric backbone structures can be modified by glycosylation and D-alanylation. WTAs promote colonisation of the host, phage binding, bacterial-cell growth, antimicrobial resistance and protection from environmental stress [92,93].

Jorge et al. showed that GlpQ from *Staph. aureus* can degrade both host phospholipids and also glycerol-type WTA molecules from coagulase-negative staphylococci (CoNS) *Staph. lugdunensis*, *Staph. capitis* and *Staph. epidermidis* [73] (Figure 3). This allowed *Staph. aureus* to proliferate under phosphorous-limited conditions. As the WTAs from CoNS were not completely degraded and their growth was not affected [73] the released phosphate might also be beneficial for GlpQ-targeted bacteria.

#### 2.3.2. Bacterial Surfaces as a Source for Sialic Acid

The surface of bacteria frequently displays sialylated oligosaccharides. Sialic acid is either synthesised de novo or acquired from the human host. It is an energy source for bacteria but can also mimic sialylated molecules on the host cell surfaces to subvert the host immune response [58,60]. The *Strep. pneumoniae* sialidase NanA can release sialic acid both from host cells and also from the surface of *H. influenzae* and *N. meningitidis*. This was proposed to decrease their attachment to host cells [94] and might also serve as carbon source for the NanA-producing *Strep. pneumoniae* (Figure 3).

#### 2.3.3. Lysis of Bacterial Cells to Release Diverse Nutrients

Different bacterial species show distinct metabolic needs and possess different biosynthetic capacities (auxotrophies and prototrophies) for amino acids or vitamins. It seems likely that species with small genomes and auxotrophic phenotypes will rely both on the host and also on other community members for essential nutrients. Interestingly, it is now recognised that bacteriocin-encoding gene clusters are widely present in human microbiomes [15]. It seems possible that secreted antibacterial compounds not only remove competitors within an ecological niche but also release essential nutrients from lysed bacteria (Figure 3). This hypothesis has hardly been investigated, but one study indicates that this might be relevant, namely *Bacillus subtilis* sacrificing some cells under nutrient-limited conditions, a phenomenon known as allolysis [95].

It is noteworthy that some species colonising the nutritionally poor nasal cavity have reduced genomes. The 1.86 Mb genome of *Dolosigranulum pigrum* suggests multiple auxotrophies for amino acids, polyamines and enzymatic cofactors. It encodes several gene clusters that putatively express diverse bacteriocins [96]. It is tempting to speculate that bacteriocin production allows *D. pigrum* to release essential nutrients from other bacterial commensals. While this needs experimental proof, it is clear that several *Corynebacterium* spp. are positively correlated with *D. pigrum* and it is assumed that *Corynebacterium* spp. either release metabolites supporting growth of *D. pigrum* (e.g., amino acids) or remove a toxic component from the medium that inhibits *D. pigrum* [96].

### 2.4. Uncharacterised Bacterial Interactions

Metagenome analysis of nasal samples from *Staph. aureus* carriers revealed that *Staph. aureus* and *C. accolens* co-occurred more often than expected by chance, while *C. pseudodiphtheriticum* was present more often in *Staph. aureus* noncarriers. In cocultivation experiments, *C. accolens* and *Staph. aureus* could support each other’s growth, whereas *C. pseudodiphtheriticum* inhibited *Staph. aureus* [97]. However, the molecular mechanisms behind these interactions are unknown.

*C. pseudodiphtheriticum* was also shown to promote host immunity to viral and bacterial infections. The application of *C. pseudodiphtheriticum* improved the outcome of infection with respiratory syncytial virus (RSV) and *Strep. pneumoniae* in a murine respiratory-tract-infection model. This was mediated by induction of T-helper-cell response [98] but the mechanistic basis of protection is unknown.

Interactions amongst members of the human nasal and respiratory tract are summarized in Table 1.

## 3. Human Diseases Altering Nutritional Composition in the Upper Respiratory Tract

Studies describing the nutritional composition of human nasal secretions have focused on healthy human volunteers [13,99]. However, little attention has been paid to the nutritional status of the volunteers. It has to be considered that the concentration of nutrients will fluctuate depending on the time passed since food intake or on underlying metabolic diseases. Glucose levels within nasal and other secretions of the respiratory tract are known to reflect those in the blood [100]. Those levels will most likely influence bacterial proliferation rates, metabolic interactions and community structures. Diabetes has a profound effect on the bacterial populations on various body sites as well as on the development of bacterial and viral infections [101]. Regarding the nasal cavity, diabetic patients have higher risks to be colonised by *Staph. aureus* than healthy individuals [102]. It is unclear whether there is a causal relationship, but it has been shown that *P. aeruginosa* numbers increase in the lungs of hyperglycemic mice compared to WT animals, a phenotype that depends on bacterial glucose acquisition [103].

In a similar fashion, other human diseases might influence the nutritional composition in the nasal cavity. For example, the intermittent interruptions in breathing of patients suffering from obstructive sleep disorder (OSD) are associated with intervals of hypoxia and hypercapnia. This will especially affect the epithelial surfaces of the respiratory tract, as air flow is significantly impaired in this condition resulting in changes in moisture, oxygen saturation [104], and eventually in the composition of the human microbiome [105]. In particular, compared to healthy individuals OSD patients display a reduced microbiome diversity with an enrichment in *Neisseria* species [106] but also in *Streptococcus*, *Prevotella* and *Veillonella* [107].

While nutrient availability and associated altered bacterial fitness might contribute to the observed microbiome alterations, more experiments are needed to support such hypotheses.

## 4. Genome-Based Metabolic Models to Predict Bacterial Interactions

Metabolic interactions amongst bacterial species are most likely to be relevant in natural communities. This is especially true for nutrient-poor environments such as the anterior nares, as well as for other human habitats such as the skin or mucosal surfaces. However, experimental evidence about relevant interactions is still rare, and when available, it is limited to a low number of model organisms that can be conveniently grown in the laboratory in single and mixed cultures under strictly defined nutritional conditions. However, in natural habitats interactions might be more complex, involving three or more species as well as fluxes of multiple metabolites between organisms. Such interactions are rarely studied experimentally, especially if solid initial hypotheses about putative interactions are lacking. In addition, some strains fail to grow under defined nutritional conditions, possibly because they would need unknown metabolites to sustain their growth.

In the age of next-generation sequencing, the scientific community has gained access to a huge number of genome sequences from different species. Analysis of genomes allows determination of the anabolic and metabolic capacities of strains. Accordingly, it is possible to model how well organisms might grow under defined nutritional conditions by determining which precursors and essential nutrients are needed and which metabolic end products would be produced. Hundreds of genome-scale metabolic models are currently available, including those of nasopharyngeal commensals and pathogens such as *Haemophilus influenzae* [108,109], *Klebsiella pneumoniae* [110], *Micrococcus luteus* [111], *Pseudomonas aeruginosa* [112], *Staphylococcus aureus* [113] and *Dolosigranulum pigrum* [114].

This offers opportunities to model the growth of an organism alone or in combination with other bacteria by suggesting metabolic interactions and fluxes. Much research has already been performed on gut microbiome communities, their relationship with the host and with each other as well as on the influence of diet on the microbiota [115,116,117,118,119,120,121,122]. One study created a workflow to investigate metabolic interactions in the nasal microbiome [123]. The next steps will be to extend modeling approaches to include three and more species and to predict primary metabolite fluxes and reciprocal support (Figure 4 blue box). Combined efforts between computational biologists and microbiologists should be able to decipher multilayered interactions within bacterial communities.

However, current metabolic models have some shortcomings that need attention. Most importantly, bacterial interactions will depend on more than just metabolic capacities. These traits can also be identified from genomic sequences and need to be considered when microbial interactions are modeled. Important factors in this regard are secondary metabolites such as siderophores. Biosynthesis genes can be identified from genome sequences. Their role in bacterial interactions is well-recognised [33]. Similarly, it is possible to predict siderophore-receptor genes in bacterial species opening avenues to incorporate siderophore-based interactions into metabolic models.

Another important trait is the production of antibacterial compounds. This is also evident from genome analysis, with bacteriocins being key players shaping bacterial communities [15]. Incorporating antimicrobial compounds into metabolic models will be challenging as their host range is often unclear. However, integration of bacteriocins will be key to improve the accuracy of the predicted interactions (Figure 4 orange box).

Finally, the relevance of secreted public goods fostering nutrient availability seems generally disregarded. Metabolic models need to integrate information regarding the presence of high molecular weight nutrients such as proteins, mucins, DNA and fatty acids as well as information about the presence of the corresponding degradative functions encoded within the genomes of the bacteria of interest. As the degradation products are accessible to the entire community it seems possible that the action of public goods rather than bacterial metabolic end products fosters reciprocal support within communities. Similarly, it needs to be considered if detoxification of host immune defense molecules such as antimicrobial fatty acids or lipids might allow growth of susceptible commensals in the context of the human host (Figure 4 green box).

Accordingly, metabolic models will be an important tool to predict microbial interactions and to suggest microbiome-editing to prevent infection. However, these approaches will only be successful if the models go beyond metabolic capacities of individual species and include the host environment as well as public goods.

## 5. Concluding Remarks

Understanding the interactions between the nasal microbiota and their influence on pathogen colonisation is important for the prevention and treatment of human infections. Due to the scarce nutrient availability in the nasal cavities, nutritional interactions between members of the nasal microbiome appear to be particularly important. Bacterial genome analyses as well as observations in vitro in isolation or in coculture showed that bacteria can use human host cells and secreted molecules such as mucins, glycans and fatty acids as nutrient sources. Furthermore, other microbiome members can act as a source of nutrients if they are lysed or surface-presented molecules are degraded. Additionally, secreted small molecules such as energy-rich fermentation products or siderophores not only benefit the producing bacterium but can also support other bacteria localised in close proximity.

Interactions occurring in vivo will involve more than just two bacterial species. This makes the importance of the known interactions difficult to judge. Moreover, several nasal bacteria have not yet been investigated in vitro as they cannot be cultured. Accordingly, future research needs to develop more sophisticated methods and experimental strategies. Genome-based metabolic models are of great help to understand under which conditions so far unculturable bacteria might grow. Simultaneously, they harbour the potential to predict interactions and metabolic fluxes between various members of complex model-communities. Such approaches are already used to investigate communities in the human gut. However, the reliability of such metabolic models will most likely increase if they incorporate the production of antibacterial compounds, secreted public goods, and information on nutrient availability in the environment. Thus, collaboration of microbiologists and bioinformaticians will be essential for further development and validation of metabolic models. Moreover, experimental strategies need to step away from too simplistic in vitro models. Future research needs the development and usage of 3D organoids that mimic the human host physiology as much as possible within the nasal cavity. Additionally, humanised animal models mimicking complex human nasal microbiomes as close as possible will be essential to realise microbiome-editing strategies.

## Figures and Tables

**Figure 1 metabolites-12-00489-f001:**
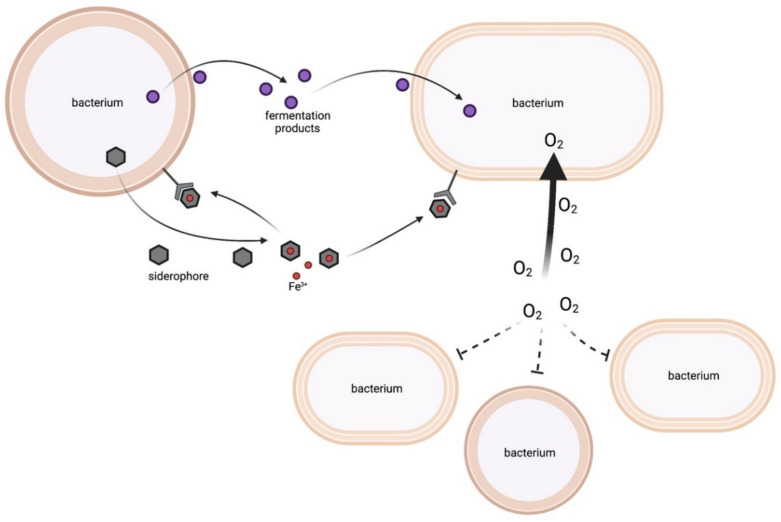
Production and consumption of small molecules. Bacteria can secrete fermentation products such as acetoin and 2,3-butanediol, which might support growth of another cohabitant bacterium. Siderophores bind iron (Fe^3+^) and can be taken up not only by the producing bacterium but also by other bacteria with a matching receptor and transporter. Oxygen consumption by one species can relieve growth inhibition of anaerobic species. This figure was created with BioRender.com.

**Figure 2 metabolites-12-00489-f002:**
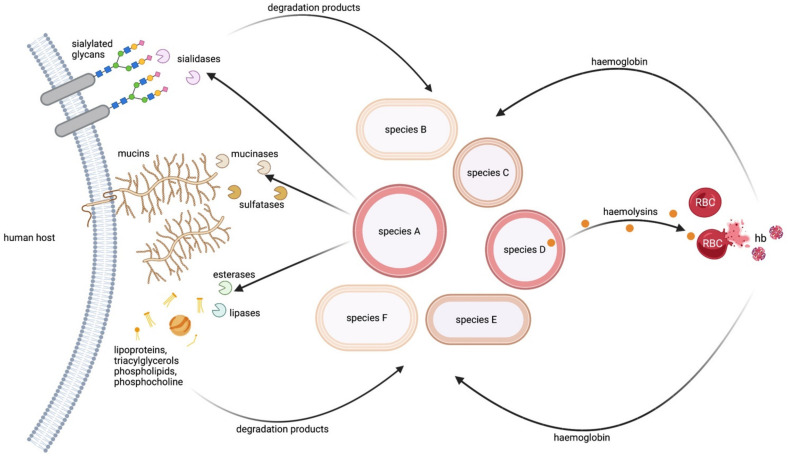
Release of nutrients from host macromolecules. Bacteria can release nutrients from different human macromolecules by secretion of degradative enzymes. Mucins (peptide chain with sugar side chains is shown) can be degraded by mucinases and sulphatases to release carbon and sulphate. Sialic acid is released from sialylated glycans (proteins modified with carbohydrates; here: blue square—N-acetylglucosamine, green circle—mannose, yellow circle—galactose, pink diamond—sialic acid) by the action of sialidases. Fatty acids, phosphate, carbon and nitrogen are released from lipoproteins, triacylglycerols, phospholipids and phosphocholine, respectively, by secretion of lipases and esterases. Lysis of red blood cells (RBCs) by haemolysins can release haemoglobin (hb). Not only the bacteria secreting the enzymes (here shown in red) but also other bacteria colonising the same niche might benefit from these nutrient sources, enhancing proliferation of the entire community. This figure was created with BioRender.com.

**Figure 3 metabolites-12-00489-f003:**
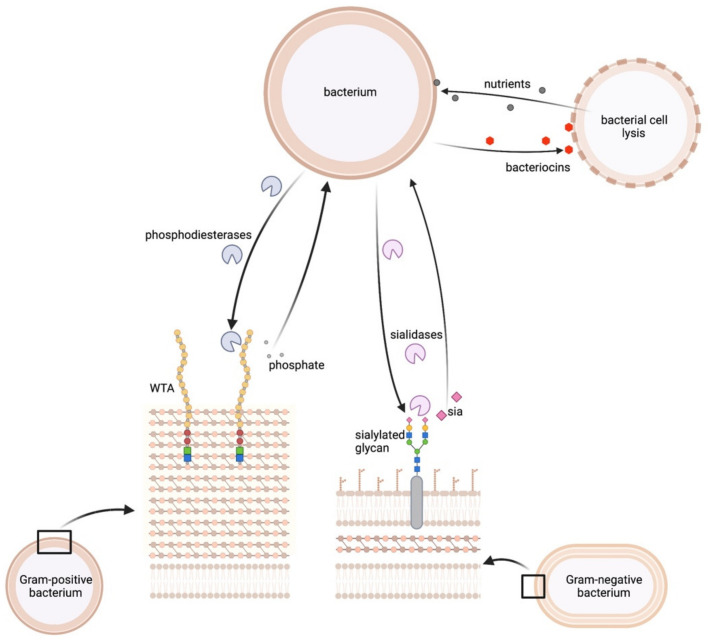
Microbiome members as a source of nutrients for other bacteria. Bacteria can degrade surface-exposed molecules of other bacteria and use them as nutrient sources. e.g., wall teichoic acids (WTA) can be degraded releasing glycerol-3-phosphate. Sialic acid (sia) can be acquired by degrading sialylated glycans. This figure was created with BioRender.com.

**Figure 4 metabolites-12-00489-f004:**
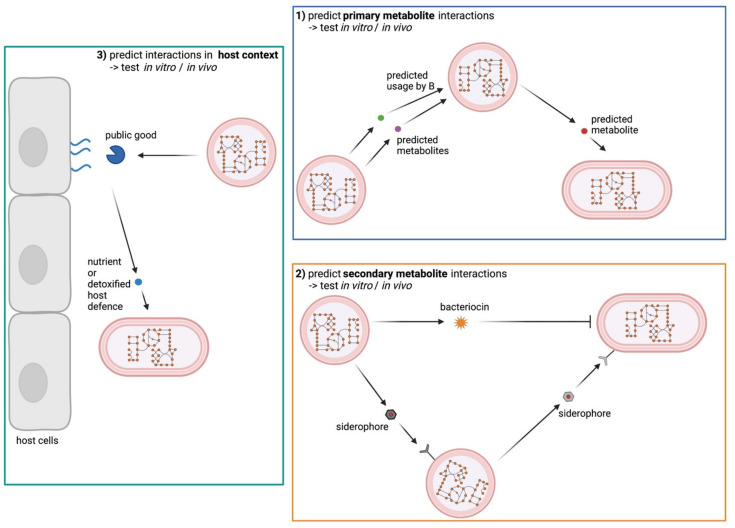
Metabolic models for prediction of bacterial interactions. Using metabolic models, primary metabolite interactions in bacterial communities can be modeled (**blue box**). Secondary metabolites such as siderophores and bacteriocins should also be regarded in predicting interactions (**orange box**) as well as public goods released from human host macromolecules (**green box**). This figure was created with BioRender.com.

**Table 1 metabolites-12-00489-t001:** Overview of metabolic interactions in the nasal, respiratory tract and skin microbiome. Metabolites or public goods, their producer, as well as species profiting or being inhibited, are indicated.

**Beneficial Interactions**				
Nutrient Source	Metabolite/Public Good	Producer of Metabolite/Macromolecule Degrading Strain	Beneficiary	Ref.
bacterial metabolism	acetoin	*Staph. aureus*	*P. aeruginosa*	[28]
bacterial metabolism	2,3-butanediol	fermenting bacteria	*P. aeruginosa* environmental microbes	[29,30]
bacterial metabolism	siderophores staphyloferrin A and B	*Staph. aureus*	*Staph. lugdunensis*	[38]
bacterial oxygen consumption	oxygen	oxygen-consuming aerobic bacteria	anaerobic bacteria	[42]
human mucins	mucin degradation products	mucin-degrading bacteria	*Staph. aureus* oral microbiome members	[47]
human mucins	mucin degradation products	*Strep. mitis*	*N. meningitidis*	[49]
human mucins	mucin degradation products/SCFAs	anaerobic communities	*P. aeruginosa*	[50,51]
human sialylated molecules	sialic acid	*Strep. pneumoniae*	*Strep. pneumoniae*	[61]
bacterial sialylated molecules	sialic acid	*H. influenzae, N. meningitidis*	*Strep. pneumoniae*	[94]
human phospholipids	glycerol-3-phosphate	*Staph. aureus*	*Staph. aureus*	[74]
WTA from CoNS	glycerol-3-phosphate	*Staph. aureus*	*Staph. aureus*	[73]
human low-density lipoproteins	fatty acids	*Staph. aureus*	*Staph. aureus*	[75]
human triacylglycerols	fatty acids	*C. accolens*	*C. accolens*	[76]
human sphingomyelin	phosphocholine	*Staph. epidermidis*	*Staph. epidermidis*	[77]
human sphingomyelin	ceramide	*Staph. epidermidis*	human host	[77]
human fatty acids	fatty acids	*P. aeruginosa*	*P. aeruginosa*	[79]
human erythrocytes	haemin & NAD+	*Staph. aureus*	*H. influenzae*	[90]
human tissue destruction	amino acids, haem	*P. gingivalis*	bacterial community	[91]
bacterial metabolism	unknown	*Corynebacterium* spp.	*D. pigrum*	[96]
**Inhibiting Interactions**				
**Nutrient Source**	**Metabolite/Public Good**	**Producer of Metabolite/Macromolecule Degrading Strain**	**Inhibited Species**	**Ref.**
bacterial metabolism	acetoin	*Staph. aureus*	*Staph. aureus*	[28]
bacterial metabolism	siderophore dehydroxynocarda-mine	*C. propinquum*	CoNS	[39]
human triacylglycerols	oleic acid	*C. accolens*	*Strep. pneumoniae*	[76]
bacterial metabolism	bacteriocins	*D. pigrum*	unknown	[96]

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
