# Peer review of "Nutritional Interactions between Bacterial Species Colonising the Human Nasal Cavity: Current Knowledge and Future Prospects"

_metabolites, 2022, doi:10.3390/metabo12060489_

Round 1
Reviewer 1 Report
- It needs to compare the references that provide similar information.
- The figure caption must provide information about all of the figure's components.
- Figure 2's caption is not specific to the figure; rather, it provides general information.
- Words need to be edited:
Line 480 that-> than
Line 493 needs-> need
5. commas should be added:
Line 480 after Unfortunately.
Line 483 after Accordingly.
Reviewer 2 Report
In this review “Nutritional interactions between bacterial species colonizing the human nasal cavity. Current knowledge and future leads”, authors summarize the concepts of nutrient release from complex host molecules and host cells as well as of intra-community exchange of energy-rich fermentation products and siderophores. I have following comments to improve the quality of review.
- How did the secretion of acetoin and 2,3-butanediol measured in fermentation? Mention it in respective paragraph.
- Make a table of beneficial and harmful microbiome released or found in nasal cavity.
- How bacterial species colonizing in nasal cavity can be helpful in supporting the host immune system against viral and bacterial?
- Authors could do a better job in discussing the genome-based metabolic models to predict bacterial interactions by making graphs or sketches and summarize them to reflect the importance of metabolic model.
Reviewer 3 Report
I have read the article by Adolf and Heilbronner with great interest. I would like to congratulate the authors for their hard work. The review focuses on the physiology of nasal microbiome. I have some comments which I hope would improve the understanding of the clinical meaning of the article.
Comments:
- What happens in diabetes with the nasal flora when there is more glucose in the nasal cavity?
- What happens in obstructive sleep apnoea with the nasal flora when the hypoxaemia would selectively influence the growth of bacteria?
- What happens in different forms of rhinitis with the nasal microbiome?
- Any article investigated nasal microbiome in vasculitis (i.e. AGPA)?
Round 2
Reviewer 1 Report
The author answered all the comments.
Reviewer 2 Report
Authors have considerably improved the manuscript. I recommend this manuscript to be published in this journal.